# Trajectory Planning for Coal Gangue Sorting Robot Tracking Fast-Mass Target under Multiple Constraints

**DOI:** 10.3390/s23094412

**Published:** 2023-04-30

**Authors:** Peng Wang, Hongwei Ma, Ye Zhang, Xiangang Cao, Xudong Wu, Xiaorong Wei, Wenjian Zhou

**Affiliations:** 1School of Mechanical Engineering, Xi’an University of Science and Technology, Xi’an 710054, China; 2Shaanxi Key Laboratory of Mine Mechanical and Electromechanical Equipment Intelligent Monitoring, Xi’an 710054, China

**Keywords:** trajectory planning, synchronous tracking, PSO, time optimization, belt sorting, multiple constraints

## Abstract

Aiming at the problems of grab failure and manipulator damage, this paper proposes a dynamic gangue trajectory planning method for the manipulator synchronous tracking under multi-constraint conditions. The main reason for the impact load is that there is a speed difference between the end of the manipulator and the target when the manipulator grabs the target. In this method, the mathematical model of seven-segment manipulator trajectory planning is constructed first. The mathematical model of synchronous tracking of dynamic targets based on a time-minimum manipulator is constructed by taking the robot’s acceleration, speed, and synchronization as constraints. The model transforms the multi-constraint-solving problem into a single-objective-solving problem. Finally, the particle swarm optimization algorithm is used to solve the model. The calculation results are put into the trajectory planning model of the manipulator to obtain the synchronous tracking trajectory of the manipulator. Simulation and experiments show that each joint of the robot’s arm can synchronously track dynamic targets within the constraint range. This method can ensure the synchronization of the position, speed, and acceleration of the moving target and the target after tracking. The average position error is 2.1 mm, and the average speed error is 7.4 mm/s. The robot has a high tracking accuracy, which further improves the robot’s grasping stability and success rate.

## 11. Introduction

Trajectory planning has always been a hot topic in robot research. An excellent trajectory planning method not only makes the robot keep good stability in the process of movement but also maximizes the benefits in time, energy, and path. In the industrial field, robots usually complete point-to-point operations [1]. This robot trajectory planning method is suitable for static target operations [2]. The main algorithms include cubic curve, quintic curve, trapezoidal curve, S curve, etc. [3]. These methods can well calculate the trajectory curve function through the trajectory boundary conditions and have high trajectory accuracy [4]. However, when the robot is tracking the moving target, these methods have great limitations, the main reason is that the constraint equations cannot meet the condition of the solution and it is difficult to obtain a definite solution.

Therefore, people began to seek new methods to achieve robot-tracking dynamic targets. Currently, the commonly used dynamic target tracking methods include the direct targeting method [5], the pyramid optimization method [6], the proportional guidance method [7], and the visual servo method [8,9,10].

The direct aiming method and point-to-point tracking method both calculate the theoretical grasping point of the target through the geometric relation of the target’s moving speed and position and then control to reach the theoretical grasping point for grasping. This method has a high requirement for the stability of belt speed. Before trajectory planning, the dynamic tracking problem is transformed into static point-to-point control. In general, the method of cosine theorem is used to calculate the dynamic target grasping point, and then it is calibrated by polynomial, B-spline curve, and other methods [11]. However, it is difficult to constrain the speed and acceleration of the manipulator in the planning, which often leads to the over-limit problem of the manipulator’s joints in pursuit of time optimization. For such problems, trajectory planning models need to be built, and then relevant constraints are added in combination with actual problems, and optimization algorithms are used to solve them, such as MOABC [12], MOHS [13], PSO [14], MOSFLA [15], PHD [16], etc.

The proportional guidance method can complete the tracking of dynamic targets, and can also complete the tracking of any target trajectory [17]. This method can have high accuracy after tracking the target and is often used for the sorting of lightweight materials in the industrial field with high sorting efficiency [18,19]. However, for fast and mass targets, there will be a great load impact after the manipulator and target reach the synchronization position when using the proportional guidance method. By analyzing the reasons, it is found that the proportional guidance method is a kind of interception tracking. When the manipulator reaches the target position, its velocity and acceleration are not matching with the target. It is because of this that velocity and acceleration are not synchronized, resulting in the failure of grasping.

Moving target tracking technology based on visual servo can be divided into two visual servo methods based on position and image according to different feedback signals [20,21]. In the former method, coordinate and pose error signals from the robot’s end-effector in Cartesian coordinate space are fed back to the robot’s controller by image processing technology to track the moving target. This method has high requirements on the visual system, especially the vision detection accuracy and target tracking speed should meet the requirements of trajectory planning [22]. The feature of the position-based visual servo control method is that the error signal and the input signal of the controller are both spatial postures, so they have clear physical significance and are relatively easy to achieve [23]. The pose error is calculated in 3d Cartesian space and needs to be reconstructed according to the image. In other words, feature points on the target are first extracted to obtain the image coordinates of the target, then the three-dimensional coordinates of the standard feature points in the camera coordinate system are calculated, and finally converted into three-dimensional coordinates in the world coordinate system according to the coordinate transformation relationship [24]. For the situation requiring attitude control, the position and attitude of the target should be calculated according to the feature points of the target in the image [25,26]. Image-based visual servoing directly uses the feature vector error signals of the target and the actuator in the image to track the target. In this method, visual servo control is generally achieved by estimating the internal and external parameters of the camera and Jacobian matrix of the robot [27], or the amount of motion of the camera is obtained by translation movement of the robot and the three-dimensional reconstruction of the target object is carried out by limiting geometric constraints to achieve control of the robot in Cartesian space [28]. Kragic, et al. [29] in Sweden used a camera to estimate the position and pose of the target relative to the camera according to the CAD model of the target and controlled the robot’s manipulator to move to a predetermined position relative to the target to grab the target. Wang, et al. [30] of The Chinese University of Hong Kong and Bishop, et al. [31] of the United States realized the speed control of the robot’s terminal in the plane through image information. Shen, et al. [32] of the Chinese University of Hong Kong proposed to adopt an ETH (Eye-to-hand) system to make the middle line of the camera’s optical axis perpendicular to the working plane of the robot’s end and construct the image’s Jacobian matrix on the working plane parallel to the camera imaging plane, which can eliminate camera parameters and realize the trajectory tracking control of the robot’s end in the working plane.

It is noted that the previous trajectory planning methods of the robot’s dynamic target tracking are mainly aimed at low-speed and low-mass targets. This kind of target can ignore the inertial load when carrying out dynamic grasping, and only by pursuing the accuracy of position can meet the operation requirements. For the big quality, fast-moving target, not only does it need to consider tracking precision but it also needs to strictly control the robot’s end before fetching the target, its position, velocity, and acceleration in sync with the target. The purpose of this planning is that dynamic tracing can introduce a fetching problem into a relatively static target capture, and in order to improve the success rate of fetching, reducing the mechanical impact load has very important significance.

The contribution of this paper is to solve the problem of fast and mass target tracking and grasping on a belt conveyor and put forward the trajectory planning method of a robot tracking moving targets under multiple constraints. Compared with other methods, this method can realize the synchronous tracking of dynamic targets and simplify the control structure. An off-line trajectory planning method of the trapezoidal curve is proposed and combined with particle swarm optimization (PSO) to achieve good tracking accuracy of the final motion parameters and target of the robot. The novelty of trajectory planning lies in that the trapezoidal curve is used to restrain the motion parameters of the robot in trajectory planning and avoid the robot from overstepping the limit. The stability of this method is proved theoretically. The method is analyzed and evaluated by simulation, and compared with existing methods. The method is also verified by a real robot system.

## 2. Models and Methods

### 2.1. Three-Stage Trajectory Planning

When planning the trajectory of the robot, it is usually necessary to consider the performance parameters of the robot. During planning, each joint of the robot is constrained, i.e., position constraint, velocity constraint, and acceleration constraint, which can be expressed as follows:(1)$$\left\{ \begin{array}{l} s_{\min}\leq s(t)\leq s_{\max}\\ |\dot{s}(t)|\leq v_{\max}\\ |\ddot{s}(t)|\leq a_{\max} \end{array}\right.$$
where *s*(*t*), $\dot{s}(t)$, $\ddot{s}(t)$ are the displacement, velocity, and acceleration at time *t*, respectively. *S*_min_ and *S*_max_ are the maximum and minimum values of the position, respectively. *v*_max_ is the extreme value of velocity and *a*_max_ is the extreme value of acceleration. The motion trajectory const *s*(*t*) raint problem is illustrated in Figure 1.

Figure 1 shows the motion speed constraint of each joint of the robot, and the initial speed is *V*_0_, corresponding to the *A*_1_ point; at *t_f_* time, when the robot tracks the target, its speed is synchronized with the target. The speed is *V_f_*, corresponding to point *A*_3_. In order to make the trajectory planning universal, the initial values of *V*_0_ and *V_f_* cannot be zero. During the movement of the robot, the maximum acceleration is *a*_max_, it can be represented by the slope of line segments *A*_1_*A*_2_, *A*_2_*A*_3_, *A*_3_*A*_4_, and *A*_4_*A*_1_. According to the acceleration constraint, it can be determined that the velocity curve should be in the parallelogram *A*_1_*A*_2_
*A*_3_*A*_4_. *L*_1_ and *L*_2_ are speed constraint curves, which can intersect *A*_1_*A*_2_
*A*_3_*A*_4_ or disjoint quadrilateral. In order to ensure that the speed curve does not exceed the limit at any time, the speed constraint curve can intersect the parallelogram *A*_1_*A*_2_
*A*_3_*A*_4_ so the speed constraint curve should be in the polygon *A*_1_*N*_1_*N*_2_*A*_3_*M*_2_*M*_1_.

For the robot, in addition to meeting the requirements of synchronization between the end-effector of the robot and the target, it should also improve efficiency to the greatest extent, which requires the robot to minimize the time in the process of completing synchronous tracking. As can be seen in Figure 1, polygon *A*_1_*N*_1_*N*_2_*A*_3_*M*_2_*M*_1_ is the safe feasible region of the robot. In order to minimize the time, the speed curve can only run along the edge contour of the feasible region. The speed curve is similar to the trapezoidal speed curve. The motion process is divided into three stages: “uniform acceleration—uniform speed—uniform deceleration”. The acceleration and deceleration process is determined by the relative position between the end of the robot and the target.

Figure 2 shows two situations in which the robot tracks a dynamic target. When tracking in the same direction, the speed curve of the robot is in a process of “uniform acceleration, uniform speed, and uniform deceleration”; in phase tracking, the robotic arm will track in reverse, then in the same direction. The speed curve of the robot is in a process of “uniform deceleration, uniform speed, and uniform acceleration”. In these two cases, the speed is generally three-stage.

In Figure 3, the three-segment velocity curve trajectory planning is an ideal trajectory planning method, but in practice, due to the inertia generated by the robot’s body and load, it is impossible to make the acceleration instantly change to zero or reach the negative value from the positive value. Of course, this is closely related to the transmission and braking mode of the robot. Even if it can be realized in a very small time, it will cause a large impact load on the body. Therefore, when planning the trajectory of each joint, considering the influence of acceleration, combined with the constraints of acceleration, velocity, and displacement, seven-segment acceleration planning is adopted, and the velocity and displacement equations are solved from the acceleration equation.

### 2.2. Seven-Segment Trajectory Planning

The belt conveyor generally operates at a constant speed, and the target can be considered as a uniform linear motion. When tracking the target, the robot first carries out variable acceleration motion with a certain acceleration. When it accelerates to the maximum acceleration, it carries out uniform acceleration motion. After uniform acceleration for a period of time, it carries out variable deceleration motion, and when the speed reaches the maximum value, it carries out uniform motion. When the end of the robot approaches the target quickly, it will move from variable deceleration to uniform deceleration to variable acceleration. When synchronized, the acceleration is zero and the speed matches the speed of the target. Acceleration planning is shown in Figure 4.

Taking *X*-axis trajectory planning as an example, *Y*-axis trajectory planning is similar. Assuming that the velocity and acceleration of the manipulator in the X direction at the initial time are 0, the initial displacement is *s* (0), and the acceleration is *j*, the kinematic equation can be established as follows:

When 0 ≤ *t* ≤ *t*_1_
(2)$$\left\{ \begin{array}{l} a(t)=Jt\\ v(t)=\frac{1}{2}Jt^{2}\\ s(t)=s\left(0\right)+\frac{1}{6}Jt^{3} \end{array}\right.$$

When *t*_1_ < *t* ≤ *t*_2_
(3)$$\left\{ \begin{array}{l} a\left(t\right)=a_{\max}\\ v\left(t\right)=v(t_{1})+a_{\max}(t-t_{1})\\ s(t)=s(t_{1})+v(t_{1})(t-t_{1})+\frac{1}{2}a_{\max}{\left(t-t_{1}\right)}^{2} \end{array}\right.$$

When *t*_2_ < *t* ≤ *t*_3_
(4)$$\left\{ \begin{array}{l} a(t)=a_{\max}-\frac{1}{2}J(t-t_{2})\\ v(t)=v(t_{2})+a_{\max}(t-t_{2})-\frac{1}{2}J{\left(t-t_{2}\right)}^{2}\\ s(t)=s(t_{2})+v_{2}(t-t_{2})+\frac{1}{2}a_{\max}{\left(t-t_{2}\right)}^{2}-\frac{1}{6}J(t-t_{2})3 \end{array}\right.$$

When *t*_3_ < *t* ≤ *t*_4_
(5)$$\left\{ \begin{array}{l} a(t)=a(t_{3})=0\\ v(t)=v(t_{3})\\ s(t)=s(t_{3})+v(t_{3})(t-t_{3}) \end{array}\right.$$

When *t*_4_ < *t* ≤ *t*_5_
(6)$$\left\{ \begin{array}{l} a(t)=-J(t_{5}-t_{4})\\ v(t)=v(t_{4})-\frac{1}{2}J{\left(t-t_{4}\right)}^{2}\\ s(t)=s(t_{4})-\frac{1}{6}J{\left(t-t_{4}\right)}^{3} \end{array}\right.$$

When *t*_5_ < *t* ≤ *t*_6_(7)$$\left\{ \begin{array}{l} a(t)=a_{\min}\\ v(t)=v(t_{5})+a_{\min}(t-t_{5})\\ s(t)=s(t_{5})+v(t_{5})(t-t_{5})+\frac{1}{2}a_{\min}{\left(t-t_{5}\right)}^{2} \end{array}\right.$$

When *t*_6_ < *t* ≤ *t*_7_
(8)$$\left\{ \begin{array}{l} a(t)=a_{\min}+J(t-t_{6})\\ v(t)=v(t_{6})+a_{\min}(t-t_{6})+\frac{1}{2}J{\left(t-t_{6}\right)}^{2}\\ s(t)=s(t_{6})+v(t_{6})(t-t_{6})+\frac{1}{2}a_{\min}{\left(t-t_{6}\right)}^{2}+\frac{1}{6}J{\left(t-t_{6}\right)}^{3} \end{array}\right.$$

In the above equations, *a*_max_ and *a*_min_ are known quantities, and the acceleration time is a constant, that is, *t*_1_ = *t*_3_ — *t*_2_ = *t*_5_ — *t*_4_ = *t*_7_ — *t*_6_. The time of uniform acceleration, uniform speed, and uniform deceleration is unknown. Trajectory planning can be carried out at a given time, which is only suitable for the fixed-point offline work of industrial robots, and the given time cannot be determined as the minimum time. For the on-line grasping of the object on the belt conveyor, it is necessary to carry out real-time planning according to the target coordinates to ensure the minimum time tracking and improve the sorting efficiency to the greatest extent. Therefore, in the track planning, the optimization algorithm is used to solve the minimum time.

## 3. Tracking Model and Solution

### 3.1. Problem Analysis

The robot starts to move when the target enters the grasping area of the robot. The time spent after the end of the robot is synchronized with the target is equal to the target motion time. When planning the trajectory of the robot, the time cannot be given accurately. Therefore, the minimum time can be solved by using the optimization algorithm, taking the time as the objective function and the trajectory of each section as the constraint. The synchronous tracking problem can be transformed into a multiple-constraint condition solution problem. There are many methods to solve this kind of problem. This paper does not focus on solving the algorithm itself but only adopts such methods to solve the mathematical model of the robot’s arm-tracking dynamic target. In this paper, the PSO algorithm is taken as an example. Taking each period of time as the optimization variable, taking the minimum total time as the optimization objective, and considering the constraints of position, speed, and acceleration in each period of time, this paper constructs the minimum time synchronous tracking model of the robot as follows:(9)$$\begin{array}{c} \text{min}\;t=\sum_{i=1}^{7}t_{i}\\ \text{Constraint}\;\text{condition}:\\ \left\{\begin{array}{l} 0<t_{i}\leq t_{\max}(i=1,2\cdots ,7)\\ a_{\min}\leq a_{i}\leq a_{\max}(i=1,2\cdots ,7)\\ v_{\min}\leq v_{i}\leq v_{\max}(i=1,2\cdots ,7)\\ s_{\min}\leq s_{i}\leq s_{\max}\\ s_{\text{obj}}=s_{\text{obj}}(t_{0})+v_{\text{obj}}t\\ s_{r}=s_{\mathrm{obj}}\\ {\text{v}}_{\text{t}7}=v_{\text{obj}} \end{array}\right. \end{array}$$
where *a*_max_ and *a*_min_ are the limit acceleration of the robot, *t_i_* is the time of each stage, and *t*_max_ is the upper limit of the time of each stage. *V*_max_, *V*_min_ is the limit speed value of the robot, *S*_min_ is the initial coordinate value of the robot’s single axis, *S*_max_ is the single axis t synchronous coordinate value of the robot, *S*_obj_ is the synchronous target coordinate value of the gangue and *V*_obj_ is the target speed of the object. In Equation (9), the first condition is the time constraint. The constraint 2–4 is the constraint on the position, speed, and acceleration of the robot, condition 5 is to calculate the position of the gangue, condition 6 is that the position coordinates of the robot are equal after tracking the gangue, and condition 7 is that the speed of the robot is synchronized with the gangue after tracking the gangue.

### 3.2. Problem Solving

Through the above analysis and modeling, we transform the trajectory planning problem of a robot tracking moving targets under multiple constraints into a single objective solution problem based on time optimization and obtain the desired trajectory through the objective function and constraints. We use particle swarm optimization to solve these problems.

PSO is initialized as a group of random particles (random solutions), and then the optimal solution is found through iteration. In each iteration, the particles update themselves by tracking two “extreme values”. The first is the optimal solution found by the particle itself. This solution is called the individual extreme value. The other extreme value is the optimal solution found by the whole population. This extreme value is the global extreme value. In the process of finding these two extreme values, the particles are updated according to Equations (10) and (11):(10)$$v_{ij}(t+1)=wv_{ij}(t)+c_{1}r_{1}[{\left(p_{ij}\right)}_{j}(t)-x_{ij}(t)]+c_{2}r_{2}[{\left(p_{gj}\right)}_{j}(t)-x_{ij}(t)]$$
(11)$$x_{ij}(t+1)=x_{ij}\left(\text{t}\right)+v_{ij}(t+1)\;\;1\leq i\leq n,1\leq j\leq d$$
where *c*_1_ and *c*_2_ are acceleration factors, the value is 2; *r*_1_ and *r*_2_ are random numbers between 0 and 1; *w* is the inertia factor, and the value is 0.5. Among them, the initial velocity and position of the particles are generated randomly and then iterated continuously according to the formula. According to Formula (9) and Figure 4, in the synchronous tracking trajectory planning of the manipulator, the unknown parameters are *t*_2_, *t*_4_, and *t*_6_, then the dimension space of the PSO algorithm can be determined to be 3, that is, the particle variable is a three-dimensional vector, and the position vector of the particle represents the possible solution. Therefore, the result of synchronous tracking trajectory planning of the manipulator can be regarded as three position vectors at the same time. Set the independent variable *X* = (*t*_2_, *t*_4_, *t*_6_) in the fitness function, and the adaptive function of the algorithm can be determined as:(12)$$f\left(X\right)=t_{2}+t_{4}+t_{6}$$


In the PSO algorithm, the number of iterations is set as 50 and the population size to 50. The specific trajectory planning process is shown in Figure 5.

When calculating the minimum time, the position, velocity, and acceleration of each time node are calculated by seven-stage trajectory planning, and the optimization is solved by constraints. The solution time of the robot’s minimum time trajectory planning method based on particle swarm optimization can be obtained through the fitness value function. In Figure 6, the x-coordinate is the number of iterations, and the y-coordinate is the fitness function value. The fitness function dimension is 3, which are particle population size, the upper limit of the parameter value, and the lower limit of the parameter value, respectively. The algorithm only needs 18 iterations to complete convergence. The algorithm can meet the system response in time.

### 3.3. Trajectory Planning Simulation

Taking the Cartesian coordinate robot as an example, this paper tracks the target on the belt conveyor, and the simulation environment is shown in the figure below (Figure 7).

In this figure, the coordinate system Og is the position coordinate of the target in the robot’s polar coordinate system, expressed in (*X_g_*, *Y_g_*, *Z_g_*), the coordinate system O is the absolute coordinate system of the robot, and the coordinate system Or is the position coordinate of the robot’s end-effector, expressed in (*X_r_*, *Y_r_*, *Z_r_*). The whole system obtains the target position through the vision system and sends the information to the robot’s controller. Considering the time complexity and space complexity of the algorithm, after receiving the target information, the robot’s controller can calculate the time and position coordinates of the target gangue entering the robot’s workspace according to the distance difference between the recognition system and the robot’s workspace. The time spent by the target gangue from the recognition area to the robot’s sorting area just provides time for the robot’s controller to carry out track planning, that is, after the robot detects the target through the recognition module, it plans the tracking track in advance, and when the target reaches the artificial working space of the machine, it controls the manipulator to track and grasp the target.

In the robot’s coordinate system, the *X*-axis is the belt running direction, the *Y*-axis is the belt width direction, and the *Z*-axis is the manipulator height direction. When the *X*-axis is in synchronous tracking, its motion equation is determined by the end of the robot and the target position. When the coordinate of the end of the robot in the *X*-axis direction is greater than the target coordinate, the robot first approaches in the reverse direction, and then carries out synchronous tracking in the same direction. When the end coordinate of the robot is less than or equal to the target coordinate, the end of the robot performs the same direction tracking, and its motion parameters are mainly determined by acceleration and velocity direction. In the *Y*-axis direction, when the robot’s end coordinate is greater than the target coordinate, the robot’s end performs negative synchronous tracking. When the end coordinates of the robot are less than or equal to the target coordinates, forward synchronous tracking is performed. According to the relative position between the end of the robot and the target, it can be divided into the following two working conditions:(13){xr≤xgxr>xg
where, *X_r_* is the *X*-axis position coordinate at the end of the manipulator, and *X_g_* is the target *X*-axis position coordinate. According to the above two working conditions, the corresponding coordinates at the end of the robot and the initial position of the target are selected for the experiment. In the experiment, the maximum speed at the end of the robot is 2.4 m/s, the maximum acceleration is 6 m/s^2^, and the belt speed is 1 m/s. The trajectory control of the manipulator is carried out by the trajectory planning method of tracking dynamic targets with multiple constraints. In the process of manipulator movement, the experimental data were counted by reading the register of the robot’s controller position, velocity, and acceleration. The target position is monitored in real time by an encoder mounted on the belt. The experimental results are shown in Figure 8 and Figure 9.

In condition 1, the target initial position coordinates are (0.3, 0.6, 0), and the initial position of the robot’s end-effector is (0.1, 0.4, 0.4). The position of the robot’s end-effector lags behind the target in the system coordinates. Therefore, when the robot carries out synchronous tracking, the *X*-axis starts to accelerate and catch up. Through the particle swarm optimization algorithm, the running time of each section of the motion trajectory is calculated, and the motion is controlled according to the calculated time. In the whole pursuit process, the *X*-axis and *Y*-axis are under linkage control. When the manipulator reaches the top of the gangue, the *Z*-axis begins to lower. At this time, the *X*-axis and the gangue remain synchronized until the manipulator completes grasping and moves up a certain safe distance to the placement point.

In condition 2, The target initial position coordinates are (0.1, 0.6, 0), and the initial position of the robot’s end-effector is (0.3, 0.4, 0.4). After receiving the sorting task, the position of the end of the manipulator is ahead of the target position. Therefore, when the manipulator performs tracking, it first approaches the target in the reverse direction, then tracks in the same direction, and finally, the two achieve synchronization. When the manipulator and the gangue are synchronized in the *X*-axis and *Y*-axis, the *Z*-axis starts to fall and execute the grasping action. In this process, the *X*-axis and *Y*-axis are required to keep synchronization with the target all the time, so as to ensure that the manipulator can stably complete the grasping action during grasping.

### 3.4. Analysis of Simulation Results

Through the simulation of dynamic target tracking of a real robot system, our method has good adaptability in both condition 1 and condition 2 mentioned above. Through the trajectory planning simulation results, it can be seen that in the robot’s tracking process, the robot’s performance can be brought into full play, and the robot’s speed and acceleration can be well controlled to ensure that the robot’s speed and acceleration do not exceed the limit in the tracking process. After the robot’s end-effector tracks the target, its position, velocity, and acceleration are consistent with the target motion state, so as to realize the target tracking synchronously.

## 4. Experimental Verification

### 4.1. Experiment

Taking the laboratory double mechanical arm coal gangue sorting robot as the research platform (as shown in Figure 10), the system is composed of a visual recognition system, belt conveyor, robot control system, robot mechanical system, and upper control system. The target is identified and positioned through the visual recognition system, and the target information is sent to the upper controller. The upper controller calculates the real-time position of the target. When the target reaches the robot’s workspace, it will carry out trajectory planning, and send the planned trajectory to the robot’s controller.

The robot adopts a rectangular coordinate structure design. Therefore, when carrying out synchronous tracking, it is necessary to consider not only the synchronous tracking problem but also the collision between the manipulator and the belt material in the tracking process. Therefore, the gate-shaped trajectory is used for trajectory planning. The sorting task of the robot is divided into four links of “rising, tracking, falling, grasping”, as shown in the track of *P*_1_*P*_3_*P*_6_*P*_8_ in Figure 11. The initial stage of rising and tracking in the *P*_1_*P*_3_*P*_6_*P*_8_ trajectory is the right-angle transition. This transition mode is easy to cause vibration and impact, which has a serious impact on the high-speed operation of the robot. Therefore, when designing the robot’s trajectory, the right-angle transition area is subject to arc transition, such as *P*_1_*P*_2_*P*_4_*P*_5_*P*_7_*P*_8_.

In the gate-shaped track, the manipulator first synchronizes with the gangue in the *X*-axis and *Y*-axis directions. After synchronization, the *Z*-axis begins to fall. After reaching the specified position, the manipulator grabs the gangue, and the *Z*-axis begins to rise after grabbing. In this case, the *X*-axis remains synchronized with the target. When the manipulator is completely separated, the placement action is executed to complete a dynamic target grabbing, according to the dynamic planning process of the coal gangue picking robot, as shown in the figure below.

During the dynamic target tracking experiment of the manipulator, the trajectory planning of the manipulator was carried out according to Figure 12. The PSO algorithm was used for time solution in three directions of the end-effector of the manipulator during planning. Due to the random position of the target on the belt, the trajectory planning time of the manipulator in each direction is calculated according to the solution method given above during trajectory planning, and then the time is substituted into the trajectory planning model. Part of the solution results using the PSO algorithm are given here, as shown in Table 1.

In robot trajectory planning, the acceleration time of the robot is quantitative, that is, *t*_1_, *t*_2_, and *t*_3_ are known. Therefore, the kinematic trajectory of the manipulator can be solved by solving the unknown quantity, and the trajectory meets the requirements of synchronous tracking of dynamic targets. In order to verify the feasibility of the method, we continue the experiment on the coal gangue sorting robot platform. Its tracking process is shown in Figure 13.

### 4.2. Result Analysis and Comparison

In order to verify the effectiveness of the robot’s dynamic target synchronous tracking method under multiple constraints proposed in this paper, we repeated PSA, DTA, and PGA methods with MATLAB, and applied them to our experimental platform. The experimental results of the above method and our method are statistically analyzed. Each algorithm uses the same parameter settings during verification. The initial coordinates of the robot are (0.1, 0.3, 0.35), the maximum speed of the robot’s end-effector is 2.4 m/s, the maximum acceleration is 6 m/s^2^, the initial coordinates of the target are (0.2, 0.1, 0), and the target moves along the X direction at a speed of 1 m/s. In order to verify the tracking effect of different methods, in the experiment, small targets are used for tracking, and the top grab is not considered in the path. The tracking tracks of different methods are shown in the figure below.

As can be seen in Figure 14, the pyramid optimization algorithm and the direct aiming algorithm are better than the algorithms in this paper and the proportional navigation algorithm in terms of tracking distance. However, when the trajectory planned by the pyramid optimization algorithm and the direct aiming algorithm approaches the target, the trajectory direction vector has a large angle with the target motion direction vector. The manipulator will cause great impact from large mass and fast-moving targets during grasping. In the process of the experiment, the impact load and grasping failure of the manipulator will be caused by too large a chamfer. The proportional guidance method has a large tracking distance in the tracking process, and there is an overtravel phenomenon in the *Z*-axis direction which causes the manipulator to hit the belt. The main reason is that the proportional coefficient needs to be optimized during the operation of the method, which increases the difficulty of the application of the algorithm. Although the method proposed in this paper is long in the tracking path, it can well control the movement direction of the manipulator at the end of the tracking track, so that the manipulator and the target can keep the same direction for cutting and grasping.

We not only care about the tracking distance but also pay more attention to whether the speed and acceleration of the manipulator and the target are consistent or infinitely close when the manipulator grabs the target after tracking the target. This can achieve relatively static grabbing and avoid impact load. Therefore, during the experiment, we specially recorded the speed and acceleration values of the four methods in the tracking process. We are more concerned about the speed and acceleration of the robot’s end-effector in the X direction. The experimental results are shown in the figure below.

In Figure 15, it can be clearly seen from the experimental results that the pyramid optimization algorithm and the direct aiming algorithm have good constraints on the speed in the whole trajectory planning process, but the only disadvantage is that after tracking the target, the speed of the robot’s end-effector is inconsistent with the target speed, which is not the result we want. In the tracking process of the proportional navigation algorithm, the speed exceeds the limit, which is very dangerous, and at the end of the trajectory, the difference between the speed of the robot’s end-effector and the target speed is larger, which causes a great impact load for the manipulator to perform the grasping action. The speed of the trajectory planned by the algorithm proposed in this paper is within the constraint range in the whole process, and when the robot’s end-effector tracks the target, its speed is consistent with the target speed, which provides good conditions for stable grasping.

In Figure 16, we can clearly see that the acceleration of the trajectory planned by the pyramid optimization method, the direct aiming method, and the proportional guidance method is seriously out of the limit, which is very dangerous for high-speed and heavy-duty robots. If the control is improper, there is a high probability of flying, which has a great impact on safety production. In the whole trajectory, our method reasonably optimizes the acceleration control without exceeding the limit and reasonably allocates the speed and deceleration time. When the robot’s end actuator approaches the target, the acceleration is close to zero to ensure that the robot’s end is relatively stationary with the target.

From the above analysis, it can be seen that the pyramid optimization algorithm, the direct aiming method, and the proportional guidance method have high tracking accuracy and speed when tracking fast and large mass targets, but when the robot’s end-effector approaches the target, its speed and acceleration cannot meet the stable grasping conditions, and it is found in the process of experiment. Using these methods, the load impact of the manipulator is caused by the mismatch between the speed and acceleration and the target many times, which causes serious damage to the manipulator and the robot. During the experiment, our method can reasonably control the position, speed, and acceleration, which makes the manipulator stay relatively stationary with the target when grasping the target, and improve the grasping stability, efficiency, and safety.

### 4.3. Error Analysis

Although the coal gangue tracking method proposed in this paper can well constrain the robot in position, speed, and acceleration, we still need to analyze the algorithm error in order to see whether it can meet the grasping accuracy. We define two errors, one is the theoretical error, the other is the actual error. The theoretical error refers to the position error calculated by the algorithm after the manipulator is synchronized with the target. The actual error refers to the position error after the manipulator and the target are synchronized in the actual operation.

When the manipulator grabs the gangue, the most important thing is the synchronization of position and speed. If there is a large dislocation between the manipulator and the target during grasping, it will lead to unstable grasping or collision, resulting in grasping failure and manipulator damage. Therefore, we focus on the position error and velocity error in the error analysis.

The theoretical error is shown in Figure 17 below. When the manipulator and the target start to perform the grasping action synchronously, the Euclidean distance between the position of the manipulator and the target position is the theoretical error of the algorithm.

In Figure 17, *Pr* is the theoretical calculation position when the manipulator is synchronized with the gangue, Pg is the gangue position, *r* is the Euclidean distance between the manipulator and the gangue, *V_r_* is the manipulator speed and *V_g_* is the gangue speed. We use *r* as the position error when the manipulator is synchronized with the gangue, then:(14)$$\left\{\begin{array}{l} \delta_{tp}=\sqrt{{\left(x_{r}-x_{g}\right)}^{2}+{\left(y_{r}-y_{g}\right)}^{2}+{\left(z_{r}-z_{g}\right)}^{2}}\\ \delta_{tv}=\left|v_{r}-v_{g}\right| \end{array}\right.$$

In order to verify the stability of the algorithm, the algorithm is run many times and the experimental results are counted. The error curve is shown in the figure below.

Through the statistical results of theoretical error, it can be seen that the position average error *δ_tp_* is in the range of 1.2 mm, and the average speed error *δ_tv_* is within 6.7 mm/s. The manipulator is a two-finger structure, and the allowable position error is 10 mm, so the theoretical calculation error meets the accuracy requirements.

In addition to verifying the calculation accuracy of the algorithm itself, it is also necessary to verify the actual error between the manipulator and the gangue in the actual work. During statistical analysis, the position of the gangue is obtained through the identification system, and then the real-time position and speed of the gangue are obtained through the belt speed measurement encoder. When the manipulator performs the grasping action, the position and speed can be obtained by the robot’s controller. The actual error calculation method is the same as the theoretical error calculation method, and then several experiments and data statistical analyses are carried out. The results are shown in Figure 18.

In Figure 19, the average position error between the manipulator and the gangue in the actual tracking and grasping *δ_ap_* is in the range of 2.1 mm, and the average speed error *δ_av_* is in the range of 7.4 mm/s. The main reason for large errors is the fluctuation of belt speed, but the error is within the range of manipulator grasping accuracy, which can ensure that the manipulator can grasp the gangue accurately.

This method has been applied in practice, with an average of 30–37 grabs per minute. The sorting robot can sort 460 kg of gangue on average per minute, which has high sorting efficiency. In order to further improve the accuracy and stability of the robot’s dynamic target capture, we will continue to deeply study the fast visual servo dynamic target tracking method on the basis of this calculation.

## 5. Conclusions

In this paper, a trajectory planning method for a robot grasping dynamic targets under multiple constraints is proposed. This method not only realizes the stable grasp of the moving target but also realizes the synchronous grasp of the target in the optimal time. This method improves the grasping stability and grasping efficiency of the robot. The conclusions are as follows:(1)Trajectory planning based on the “seven segment” acceleration curve can effectively improve the stability of the robot’s operation and avoid load impact. By constraining the position, speed, and acceleration, it can effectively prevent the driving overrun of the robot and improve the safety of the robot;(2)The dynamic target tracking algorithm does not only realize the accurate tracking of the robot, but also ensures the grasping stability, and solves the problem of dynamic grasping of high-speed and high-quality targets to a certain extent. After the manipulator is synchronized with the dynamic target, the position error is 2.1 mm and the speed error is 7.4 mm/s;(3)The trajectory planning method proposed in this paper can give full play to the driving performance of the robot, and make the robot quickly synchronize the moving target in a stable motion;(4)This method is not only suitable for the synchronous tracking of uniform linear moving targets but also for the point-to-point grasp trajectory optimization of industrial robots;(5)Although the proposed algorithm can realize trajectory planning of manipulator tracking dynamic targets, the real-time performance and accuracy of the algorithm need to be further improved. In future work, the computational time complexity and accuracy will be further studied to improve the robot’s grasping efficiency.

## Figures and Tables

**Figure 1 sensors-23-04412-f001:**
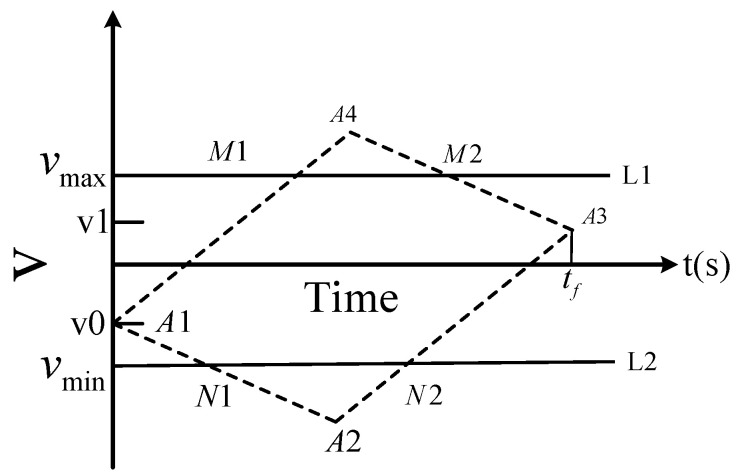
Speed constraints.

**Figure 2 sensors-23-04412-f002:**
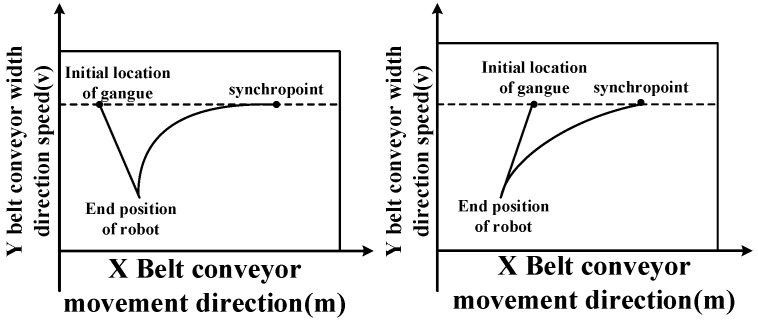
Schematic diagram of phase tracking and co-direction tracking.

**Figure 3 sensors-23-04412-f003:**
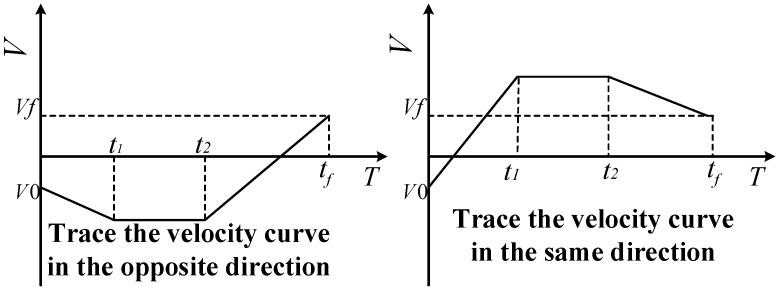
Speed planning curve.

**Figure 4 sensors-23-04412-f004:**
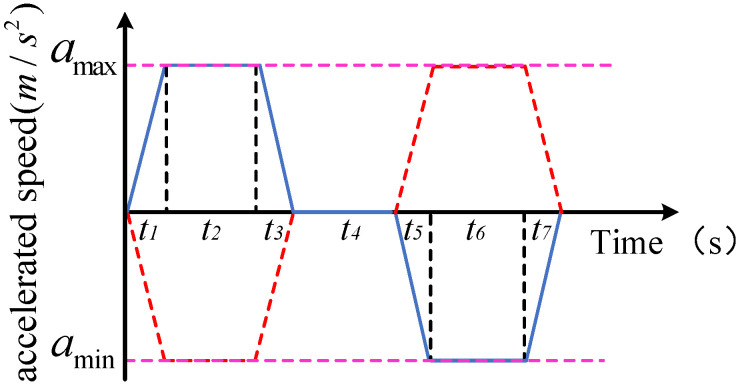
Acceleration constraint curve.

**Figure 5 sensors-23-04412-f005:**
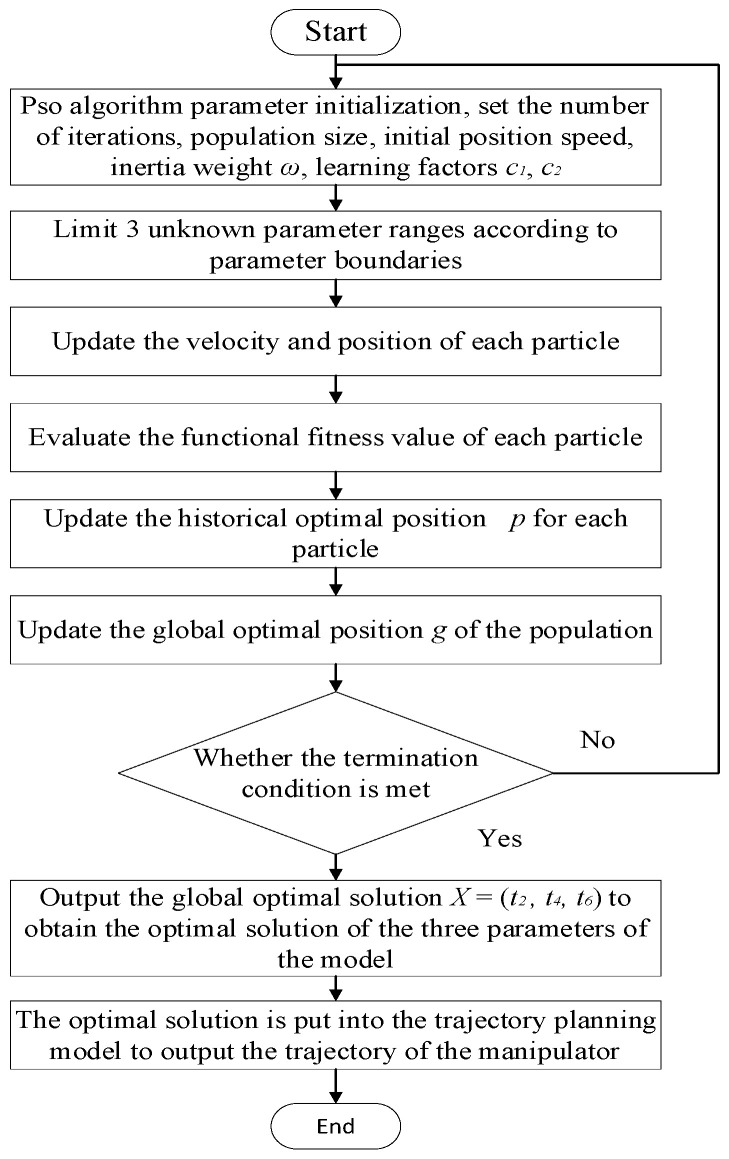
Robot trajectory planning solution process based on the PSO algorithm.

**Figure 6 sensors-23-04412-f006:**
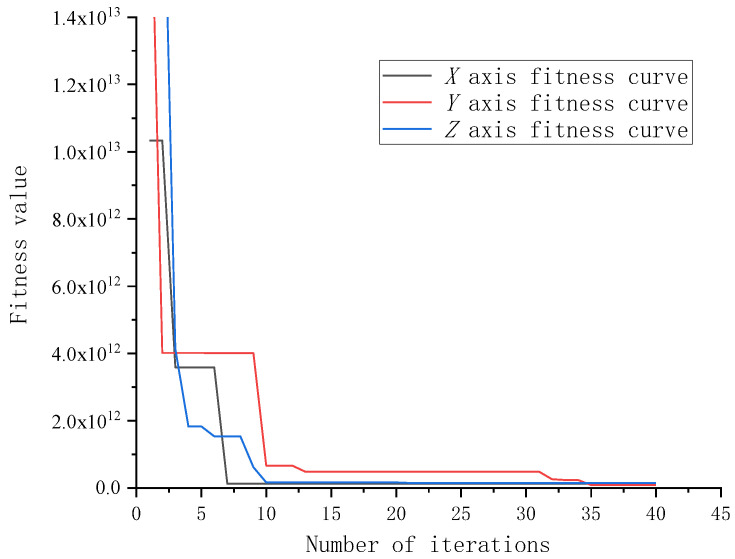
Fitness curve.

**Figure 7 sensors-23-04412-f007:**
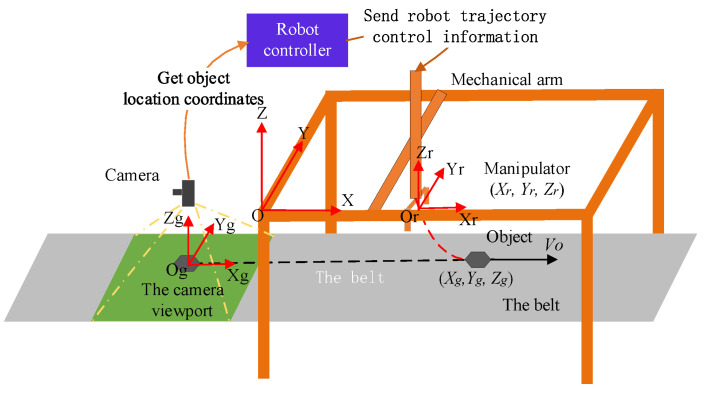
Simulation environment of belt conveyor.

**Figure 8 sensors-23-04412-f008:**
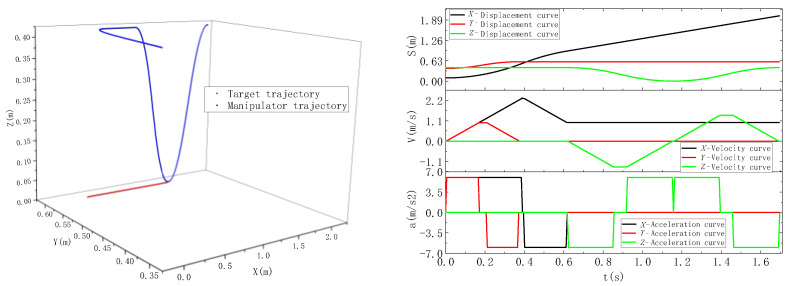
Robot tracking track under working condition 1.

**Figure 9 sensors-23-04412-f009:**
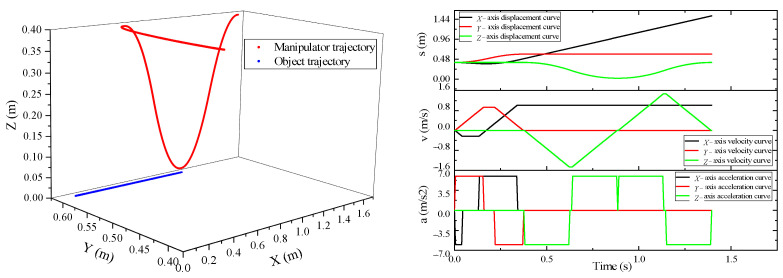
Robot tracking track under condition 2.

**Figure 10 sensors-23-04412-f010:**
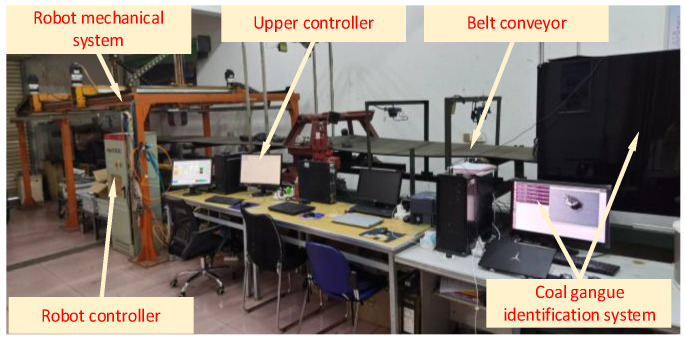
Coal gangue sorting robot with double mechanical arms.

**Figure 11 sensors-23-04412-f011:**
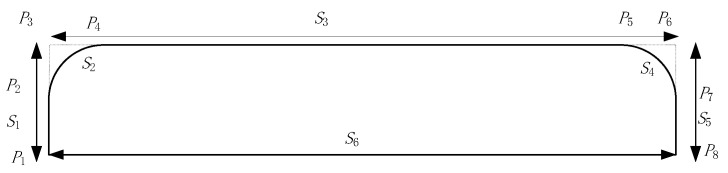
Point-to-point gate glyph trajectory.

**Figure 12 sensors-23-04412-f012:**
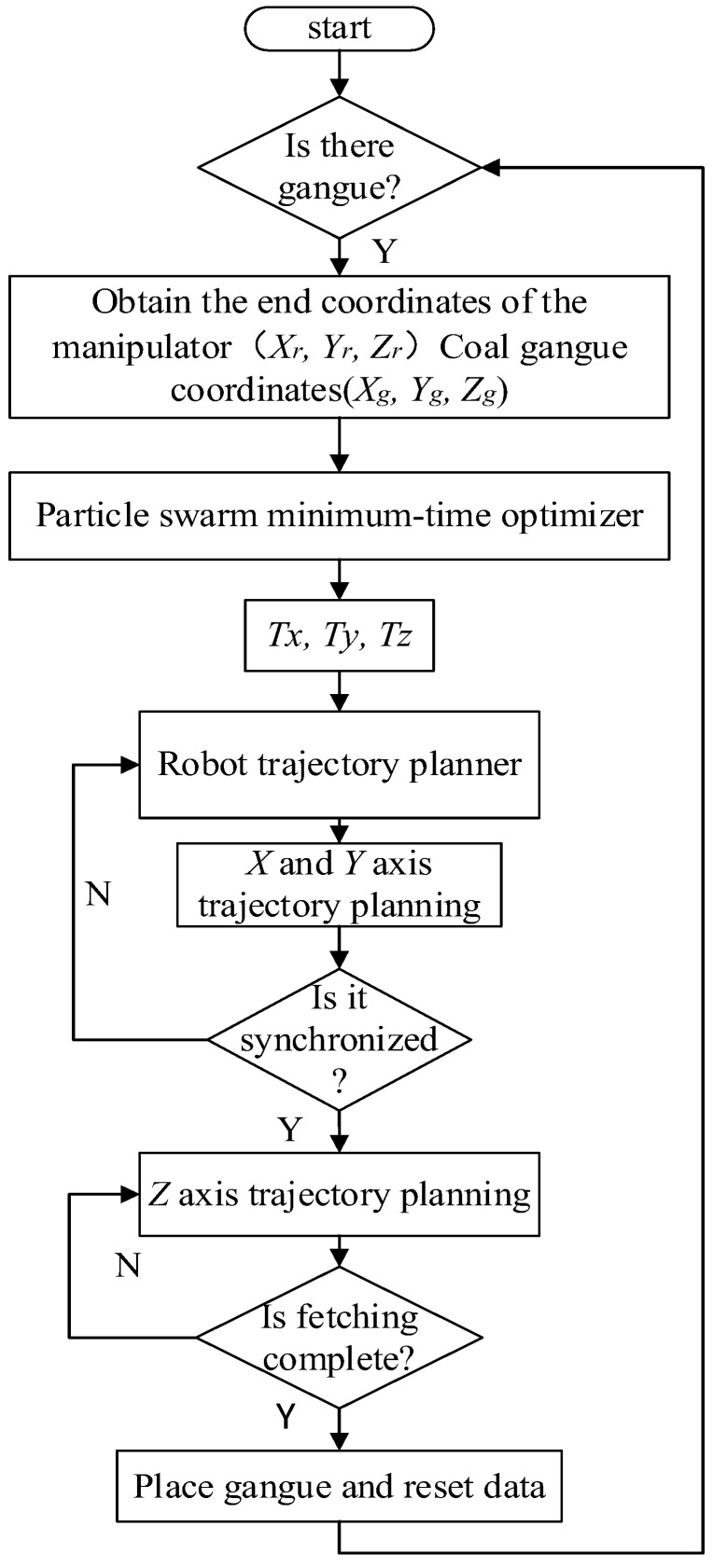
Flow chart of the robot’s synchronous tracking algorithm.

**Figure 13 sensors-23-04412-f013:**
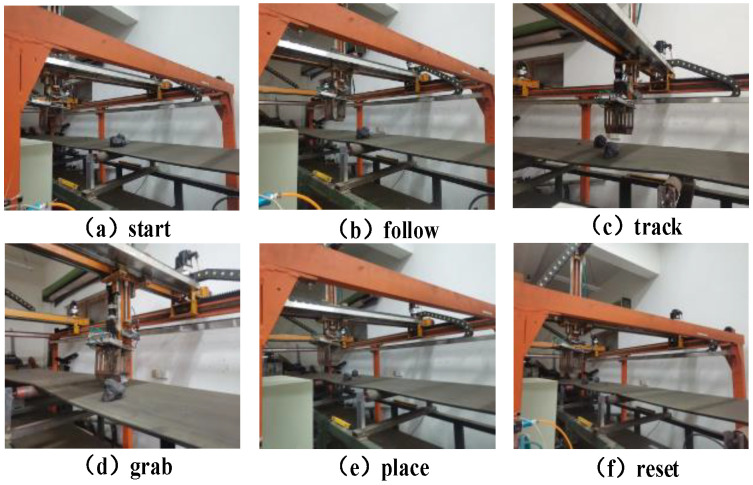
Process of synchronous tracking and grasping target gangue by the manipulator.

**Figure 14 sensors-23-04412-f014:**
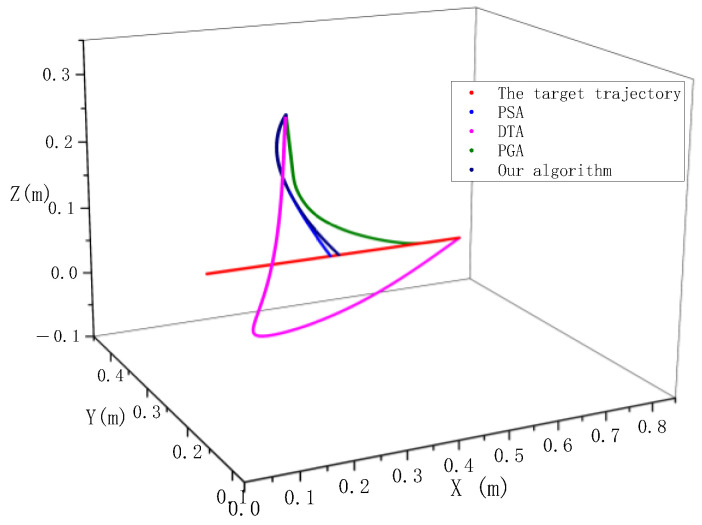
Comparison of tracking trajectories of different algorithms.

**Figure 15 sensors-23-04412-f015:**
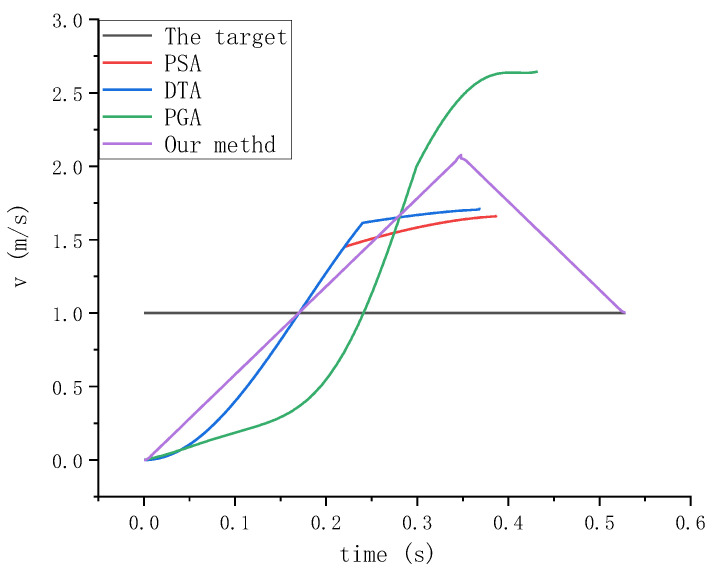
Velocity curves of different algorithms in X direction.

**Figure 16 sensors-23-04412-f016:**
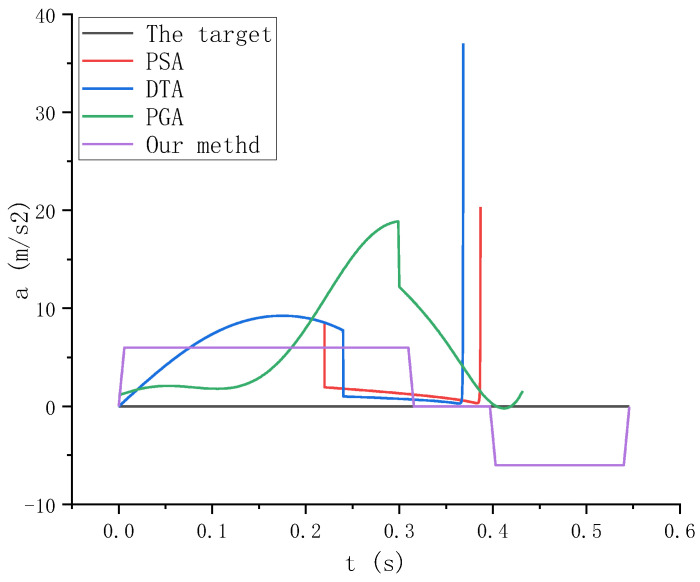
Acceleration curves in the X direction of different algorithms.

**Figure 17 sensors-23-04412-f017:**
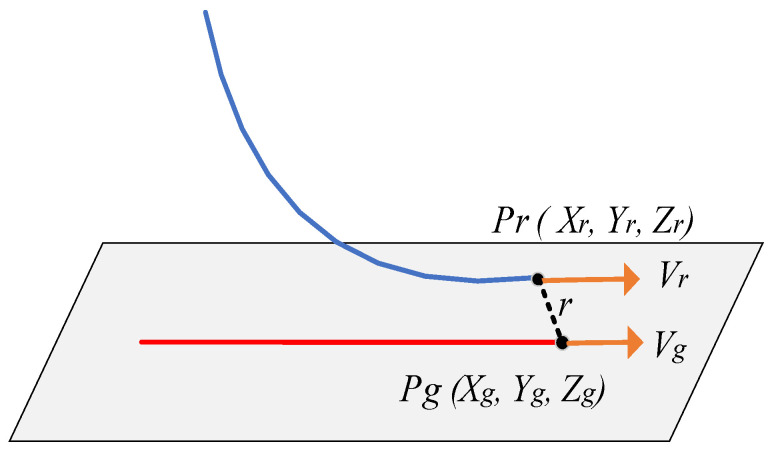
Error schematic diagram.

**Figure 18 sensors-23-04412-f018:**
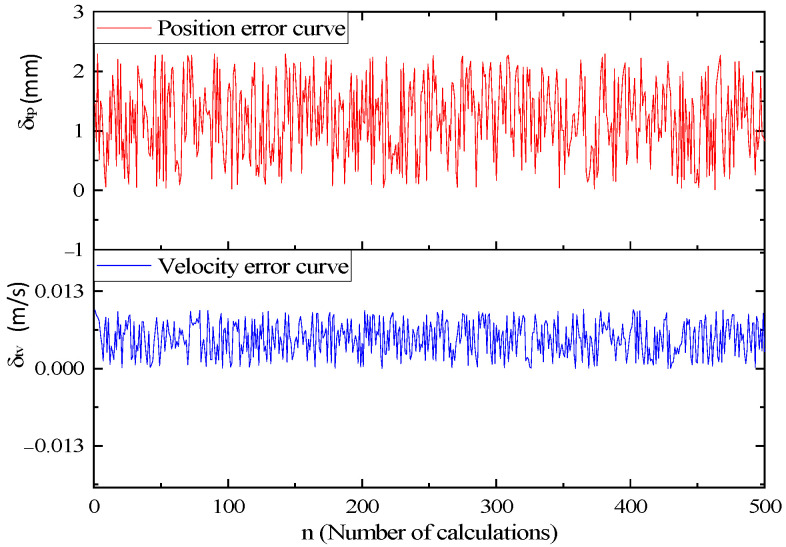
Theoretical error curve.

**Figure 19 sensors-23-04412-f019:**
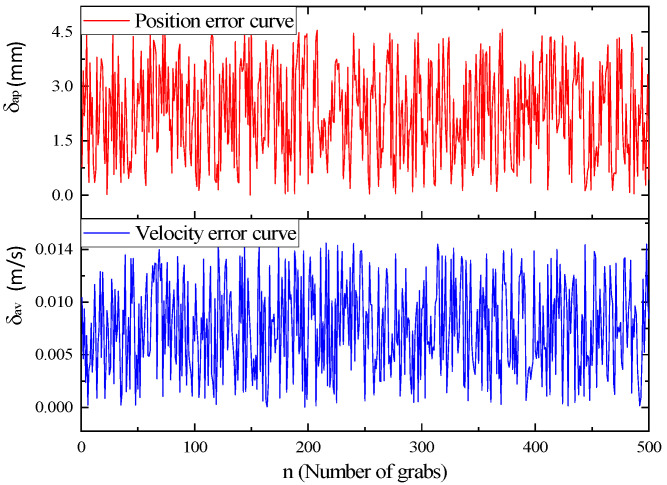
Actual error curve.

**Table 1 sensors-23-04412-t001:** Solution results based on the PSO algorithm.

Target Initial Position	Initial Position of the Manipulator	Coordinate Axis	*t*_2_ (s)	*t*_4_ (s)	*t*_6_ (s)
(0.3, 0.6, 0)	(0.05, 0.4, 0.4)	*X*	0.362	0.086	0.201
		*Y*	0.174	0.004	0.178
		*Z*	0.237	0.053	0.239
(0.3, 0.6, 0)	(0.3, 0.4, 0.4)	*X*	0.244	0.050	0.078
		*Y*	0.151	0.029	0.150
		*Z*	0.210	0.065	0.211
(0.1, 0.6, 0)	(0.3, 0.4, 0.4)	*X*	0.007	0.076	0.171
		*Y*	0.153	0.075	0.156
		*Z*	0.213	0.045	0.212

## Data Availability

Data available on request due to restrictions, e.g., privacy or ethical. The data presented in this study are available on request from the corresponding author. The data are not publicly available due to data related to research project confidentiality.

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
