# Peer review of "Trajectory Planning for Coal Gangue Sorting Robot Tracking Fast-Mass Target under Multiple Constraints"

_sensors, 2023, doi:10.3390/s23094412_

Round 1

Reviewer 1 Report

1.       In abstract, the background and meaning of research are not mentioned. Please figure out in the first to second sentences. Then, please introduce the existing problems in this research.

2.       In abstract, the overall structure of abstract should be adjusted. From the view of followings: 1. Background, existing problems 2. Proposed method 3. Details of method 4.experiment result. Now, I have not seen any data of result.

3.       The main contribution of this paper is suggested to move to the end of introduction, not mentioned in the abstract.

4.       In introduction, there are large of swarm intelligence algorithms related to solve the multi-objective problems. These related works are not mentioned. And why do you choose PSO algorithm to optimize the tracking route? Please give out the reasons.

5.       Please analyze and add them in related works, such as MOABC, MOHS,MOSFLA. Some related works are suggested to listed, such as follows

[1] An ant colony optimization algorithm with adaptive greedy strategy to optimize path problems. Journal of Ambient Intelligence and Humanized Computing, 2021.

[2] Harmony Search Method with Global Sharing Factor Based on Natural Number Coding for Vehicle Routing Problem. Information, 2020

[3] Capacitated vehicle-routing problem model for scheduled solid waste collection and route optimization using PSO algorithm. Waste Manag, 2017

[4] Application of adaptive grid-based multi-objective particle swarm optimization algorithm for directional drilling trajectory design. Geoenergy Science and Engineering, 2023

[5] Equal probability embedded cubature particle PHD filter algorithm in multi-target tracking. Proceedings of the Institution of Mechanical Engineers, Part I: Journal of Systems and Control Engineering, 2023

6.       In the 3.1 Problem analysis part, the equation 9 is the key of the multi-objective optimization problem. Unfortunately, in Equation 9, there is a simplified single objective function which using the method of average weight. Apparently, it is not belong to the multi-objective optimization field strictly. And the coefficients of ti in summary equation are not mentioned.

7.       The multi-objective optimization problem is descripted so less. The novelty and improvement of this paper appears so weak.

8.       In section 3.2 Problem solving, please figure out how the initial parameters of PSO algorithm represent the robot parameters. And figure out the fitness function of PSO.

9.       The Fig.5 Flow chart of PSO is wrong. The author thought the equation 9 is a multi-objective optimization problem. And the fig 5 shows it is a single objective problem. In fact, the equation 9 is not a multi-objective optimization problem. Please revise the error.

10.    The time complexity and space complexity are not analyzed.

11.    In experiment, any of optimization data result from multi-objective PSO are not mentioned.

12.    In Conclusion section, it is too short, not mentioned optimization results.

13.    In the end, the flaw of the paper and future work are not given.

Author Response

I would like to thank the reviewers for their valuable suggestions, which are of great help to my paper. I have made modifications one by one according to the reviewer's suggestions. The changes have been highlighted in blue in the text. See the attachment for details.

Reviewer 2 Report

The article contains interesting new results. However, there are a number of remarks that need to be addressed:

1. Show curves L1 and L2 in Figure 1.

2. What is shown in Figure 2? There is no reference in the text to Figure 2 and no commentary on it.

3. The abscissa axes are not labeled in Figure 3.

4. The axes in Figure 4 show the dimensionality. However, there are no numerical values in Figure 4. This is incorrect.

5. What is the difference between a(t1) and amax in formula (3)?

6. For the second and third equations of system (5) to be correct, the first equation must be written: a(t)=a(t3)=0.

7. How is the velocity v(t3) in the system of equations (5) different from the velocity v(t4) in the system of equations (6)?

8. It is not clear what the unnumbered equation before the system of equations (9) expresses. The min sign should be in the left and right parts of the equation. Or absent in both left and right parts of the equation.

9. Equation (5) of the system of equations (9) can be correctly written as follows: sobj(t)=sobj(t0) + vobj(t).

10. Equations (6) and (7) of the system of equations (9) are incomprehensible. Explain, does capture occur immediately at the moment when the speed and position of the robot and the target are synchronized? Or does the robot follow the target for some time after synchronization?

11. In general case amax and amin are different in modulo? Can we not write amin=-amax?

12. In the right-hand side of equation (11) instead of xij we should write xij(t).

13. What is the dimensionality of the ordinate axis in Figure 6? At what values of the parameters are the results obtained in Figure 6. Explain.

14. Why does the equation in front of figure 8 again have the number 10? Equations (10) and (11) already existed before.

15. Tell me more about the experiment. How were the results obtained in figure 8? Write down the equations of the model. Present the numerical values of the parameters. 

16. Why did the acceleration jumps in Figure 8 (lower block, the black curve at the beginning and the green curve just under 1 s) have no effect on the velocity graphs? What caused these jumps?

17. State the dimensions of the ordinate axes in Figures 18 and 19.

Author Response

Thank the reviewers for their valuable comments. The suggestions put forward are of great help to my thesis. I have modified the paper as required by the reviewer. The changes are highlighted in blue. See the attachment for details.

Round 2

Reviewer 1 Report

Accept

Author Response

Thank the reviewers for their guidance on the paper. I have corrected the language problems in this article.

Reviewer 2 Report

I am satisfied with the work done by the authors. I believe that the paper can be published.

Author Response

(The authors gave the same response as above.)
